# Absence of Dx2 at *Glu-D1* Locus Weakens Gluten Quality Potentially Regulated by Expression of Nitrogen Metabolism Enzymes and Glutenin-Related Genes in Wheat

**DOI:** 10.3390/ijms21041383

**Published:** 2020-02-18

**Authors:** Lijun Song, Liqun Li, Liye Zhao, Zhenzhen Liu, Tingting Xie, Xuejun Li

**Affiliations:** College of Agronomy and State Key Laboratory of Crop Stress Biology in Arid Areas, Northwest A&F University, Yangling 712100, China; lijunsongNWAFU@163.com (L.S.); liliqun@nwsuaf.edu.cn (L.L.); liyezhao5526@163.com (L.Z.); zhenzhenliu1717@163.com (Z.L.); xietingting95@163.com (T.X.)

**Keywords:** glutenin-related gene, high-molecular-weight glutenin subunit Dx2, nitrogen metabolism enzyme, protein body, wheat gluten quality

## Abstract

Absence of high-molecular-weight glutenin subunit (HMW-GS) Dx2 weakens the gluten quality, but it is unclear how the absence of Dx2 has these effects. Thus, we investigated the gluten quality in terms of cytological, physicochemical, and transcriptional characteristics using two near-isogenic lines with Dx2 absent or present at *Glu-D1* locus. Cytological observations showed that absence of Dx2 delayed and decreased the accumulation of protein bodies (PBs), where fewer and smaller PBs formed in the endosperm. The activity and gene expression levels of nitrogen assimilation and proteolysis enzymes were lower in HMW-D1a without Dx2 than HMW-D1p with Dx2, and thus less amino acid was transported for protein synthesis in the grains. The expression pattern of genes encoding Glu-1Dx2+1Dy12 was similar to those of three transcription factors, where these genes were significantly down-regulated in HMW-D1a than HMW-D1p. Three genes involving with glutenin polymerization were also down-regulated in HMW-D1a. These results may explain the changes in the glutenin and glutenin macropolymer (GMP) levels during grain development. Therefore, we suggest that the lower nitrogen metabolism capacity and expression levels of glutenin synthesis-related genes in HMW-D1a accounted for the lower accumulation of glutenin, GMP, and PBs, thereby weakening the structural‒thermal properties of gluten.

## 1. Introduction

Wheat grain proteins are classified as albumins, globulins, gliadins, and glutenins based on their solubility properties. Albumins and globulins are soluble proteins comprising various enzymes and inhibitors, and they have important structural and metabolic functions during grain-filling [1]. Gliadins and glutenins are storage proteins that account for 80–85% of the total wheat flour proteins, and they interact to form gluten [2]. The monomeric gliadins contribute mainly to the viscosity and extensibility of wheat dough, and the polymeric glutenins to the elasticity and dough strength [3]. Glutenins are subdivided into low and high-molecular-weight glutenin subunits (LMW-GS and HMW-GS, respectively) [4]. HMW-GSs account for only 20% of the total glutenin fractions in terms of quantity, but they make major contributions to the strength and elastic properties of dough [5].

HMW-GSs are specified by three homologous loci (*Glu-A1*, *Glu-B1*, and *Glu-D1*) located on the long arms of group 1 chromosomes [6,7]. Each *Glu-1* locus contains two tightly linked genes encoding x- and y-type subunits. However, common wheat cultivars produce three to five HMW-GSs due to the silencing of certain genes. The x- and y-type HMW-GSs differ from each other in terms of the numbers and distributions of cysteine residue within their domains [7]. These cysteine residues are involved in intermolecular disulfide bonding during the formation of larger polymeric proteins, and they play important roles in the functions of HMW-GSs [5,7]. Several studies have reported that variations in the numbers of HMW-GSs account for variations in the rheological properties of wheat dough [4,7]. In our previous study, we used near-isogenic lines (NILs) to investigate the effects of HMW-GS variations at the Glu-D1 locus on the microstructure of gluten and rheological properties of dough, where the different contributions of HMW-GSs to the microstructure and rheology were ranked as: Dx5+Dy10 > Dx2+Dy12 > Dy12 [8,9]. In addition, the absence of Dx2 delayed glutenin polymerization during grain development and affected the gluten quality [8]. However, how the absence of Dx2 affects the polymerization of glutenin and the quality of gluten remains unclear.

*HMW-GS* genes are specifically expressed in the endosperm and they have similar expression patterns [10,11]. Regulation of the expression of *HMW-GS* occurs primarily at the transcriptional level, where it involves cis-acting motifs in the HMW-GS promoters and trans-acting transcription factors (TFs) [4]. The *HMW-GS* gene promoter consists of five common motifs recognized by the corresponding TFs. The storage protein activator (*SPA*) regulates *HMW-GS* gene expression by recognizing the GCN4-like motif in the *HMW-GS* promoter, and the prolamin-box binding factor (*PBF*) binds to the prolamin-box [12]. TaFUSCA3 interacts with *TaSPA* to activate the HMW-GS gene [13]. Gibberellin-response myeloblastosis (*TaGAMyb*) activates the expression of HMW-GS genes by recruiting histone acetyltransferase encoded by the *TaGCN5* gene during wheat endosperm development [10]. However, the effects of these TFs on glutenin accumulation in wheat with different HMW-GSs according to the *Glu-D1* locus have not been elucidated. Moreover, some genes that are not directly involved in the expression of storage protein genes may influence the biosynthesis of storage proteins, such as those encoding glutamine synthetase (GS) and glutamate pyruvate transaminase (GPT) [14]. Overexpression of the genes encoding GS and GPT enhances amino acid metabolism [15,16]. Weber et al. [17] suggested that the use efficiency of amino acids during protein translation can directly affect the amounts of storage proteins. However, the relationship between nitrogen remobilization and the synthesis of storage proteins requires further investigation in wheat, especially according to the variations in HMW-GSs.

Gluten proteins are formed and deposited in endosperm organelles, protein bodies (PBs) derived from the rough endoplasmic reticulum, and protein storage vacuoles during wheat seed development [18]. Protein-folding and assembly occur in the lumen of the endoplasmic reticulum with the assistance of complex interactions. Peptidyl-prolyl cis-trans isomerase (PPIase) acts as a protein-folding catalyst to accelerate isomerization of the peptidyl-prolyl bond and the reorganization of disulfide bonds [19]. Small ubiquitin-related modifier 1 (SUMO1) is involved in the transport of proteins from the nucleus to the cytoplasm, and it contributes to the stability of proteins [20]. PPIase and SUMO1 interact to facilitate the formation of protein polymers [21]. Protein disulfide isomerase (PDI) catalyzes the formation of disulfide bonds, which are regarded as the determinant bridges during protein polymerization [22]. The high level expression of genes encoding PPIase, SUMO1, and PDI enhances the folding of storage proteins during grain-filling, thereby leading to the accumulation of more glutenin macropolymers (GMPs) [11,21,23]. HMW-GSs contribute to glutenin polymerization via intra-/intermolecular disulfide bonds and hydrogen bonds [4], and the effects of single HMW-GSs on glutenin polymerization have been reported [8]. However, the effects of the absence or presence of single HMW-GSs on the folding and assembly of glutenin during grain development are unclear.

In the present study, we employed wheat NILs with the absence/presence of Dx2 at the *Glu-D1* locus to further investigate the effects of Dx2 on the quality of gluten. In particular, we analyzed the accumulation of storage proteins, the activities and gene expression levels of nitrogen metabolism enzymes, and the expression of genes related to glutenin synthesis and folding during grain development. Our results provide insights into how HMW-GS Dx2 absence weakens the quality of wheat.

## 2. Results

### 2.1. Analysis of Structural‒Thermal Properties of Gluten

The gluten from HMW-D1p had a higher disulfide bond concentration and lower values for the sulfhydryl groups and surface hydrophobicity compared with the gluten from HMW-D1a (Figure 1A,B). The results also showed that compared with HMW-D1a, the gluten from HMW-D1p had higher values for the denaturation peak temperature, enthalpy of thermal transition, and degradation temperature, but lower weight loss (Figure 1C‒F). The gluten microstructure of dough was more compact and denser for HMW-D1p than HMW-D1a (Figure 1G). The quantitative analysis of the microstructure (Table 1) showed that compared with HMW-D1a, HMW-D1d exhibits significantly higher values in protein area, protein junctions, junction density, total protein length, lacunarity, and branching rate, whereas it exhibits lower endpoints and endpoint rate. These findings suggest that HMW-D1a produced gluten with weaker structural‒thermal stability than HMW-D1p.

### 2.2. Dynamic Accumulations of Protein Fractions in Grains during Grain Development

The protein fractions obtained from HMW-D1a and HMW-D1p are shown in Figure 2. The total protein contents varied slightly but there were no significant differences between HMW-D1a and HMW-D1p during the entire grain-filling stage, except at 28 days after anthesis (DAA) (Figure 2A). The albumin and globulin contents decreased gradually in the early filling stages but did not change subsequently (Figure 2B), whereas the opposite trend was found for gliadin during the grain-filling stage (Figure 2C). The albumin, globulin, and gliadin contents did not differ significantly in HMW-D1a and HMW-D1p. The glutenin contents increased slightly in the early filling stage, but they accumulated rapidly from 22 to 34 DAA (Figure 2D). The fastest accumulation of GMP was also gained from 22 to 34 DAA, but the increase rate of GMP in HMW-D1p is higher than that in HMW-D1a (Figure 2E). The results suggest that the absence of Dx2 delayed and decreased the polymerization of glutenin.

### 2.3. Accumulation of PBs in Endosperm during Grain Development

To clarify the effects of the absence of Dx2 on the accumulation of storage proteins, cytological observations were performed using the endosperm from HMW-D1a and HMW-D1p. Figure 3 shows that numerous PBs formed during grain development. At 10 DAA, the endosperm cells contained a low abundance of small starch granules, and no PBs were observed in HMW-D1a (Figure 3A) whereas a few PBs were present in the HMW-D1p endosperm cells (Figure 3D). At 13 DAA, some PBs appeared in the HMW-D1a endosperm, but they were smaller than those in HMW-D1p (Figure 3B,E). More abundant and larger PBs accumulated in the endosperm cells of HMW-D1p than HMW-D1a at 16 DAA (Figure 3C,F). Additionally, the decrease rate of storage protein at 10, 13, and 16 DAA caused by the absence of Dx2 were 21.65%, 20.60%, and 10.31% (Figure 2C,D), respectively. Therefore, it can suggest that the absence of Dx2 has negative effect on the storage protein accumulation in the endosperm.

### 2.4. Changes in GS and GPT Activities and Levels of Amino Acids in Flag Leaves and Grains during Grain Development

The GS and GPT activities were determined in the flag leaves and grains during grain-filling, as well as the amino acid contents (Figure 4 and Figure 5). The GS activities in the flag leaves and grains exhibited similar patterns during grain-filling (Figure 4A,B), whereas the GPT activity patterns differed in the flag leaves and grains during grain development (Figure 4C,D). The activities of the two enzymes followed similar patterns in HMW-D1p and HMW-D1a, but the values in HMW-D1p were higher than those in HMW-D1a at the same development stage. Moreover, the maximum amino acid contents in the flag leaves occurred at 10 and 13 DAA for HMW-D1p and HMW-D1a, respectively, and the contents were higher in HMW-D1p than HMW-D1a (Figure 5A), which was also the case for the amino acid contents in the grains (Figure 5B). These findings indicate that the absence of Dx2 delayed and decreased the activities of GS and GPT, thereby leading to the availability of lower amounts of amino acids for subsequent protein synthesis.

### 2.5. Expression of Genes Related to Nitrogen Metabolism during Grain Development

The expression levels of TaGS1, TaGS2, TaAlaAT (alanine aminotransferase), TaWCP2 and TA.610216 related to nitrogen metabolism are shown in Figure 6. The expression patterns of these genes differed in the flag leaves and grains, where the expression levels were higher in HMW-D1p than those in HMW-D1a during the same stages. The expression level of TaGS1 in the flag leaves increased from 0 DAA to 7 DAA for HMW-D1p and from 0 DAA to 10 DAA for HMW-D1a, before the expression levels declined in both varieties (Figure 6A). The maximum expression level of TaGS2 in the flag leaves occurred at 4 DAA for HMW-D1p and at 7 DAA for HMW-D1a (Figure 6B). The expression level of TaAlaAT exhibited similar patterns in the flag leaves from both varieties, where the highest expression levels occurred at 10 DAA (Figure 6C). The expression levels of TaWCP2 and TA.610216remained stable in the flag leaves between 0 DAA and 22 DAA, and then increased rapidly until 28 DAA, where the levels were 4‒5 fold higher than those in the early filling stages (Figure 6D,H). The expression levels of TaGS1 and TaGS2 in the grains were highest at 4 DAA, before declining rapidly in HMW-D1p and HMW-D1a (Figure 6E,F). The expression level of TaAlaAT was high in the grains before 16 DAA for HMW-D1p, but the maximum expression level was not reached until 16 DAA for HMW-D1a (Figure 6G). These results indicate that the capacity for nitrogen metabolism was lower in HMW-D1a than HMW-D1p.

### 2.6. Transcription Levels of Genes Related to Glutenin Synthesis and Polymerization during Grain Development

Figure 7 shows the expression levels of genes related to the synthesis (Glu-1Dx2+1Dy12) and aggregation (TaPDIL2-1, TaPPIase, and TaSUMO1) of glutenin and TFs (TaPBF, TaSPA, and TaGAMyb) during grain development (4‒28 DAA). The expression levels of Glu-1Dx2+1Dy12 in HMW-D1a and HMW-D1p gradually increased from 4 to 13 DAA, before reaching a peak at 16 DAA and then declining markedly (Figure 7A). The expression levels of three genes comprising TaGAMyb, TaPPIase, and TaSUMO1 had similar patterns in the same stages and they peaked at 13 DAA in both lines (Figure 7B,E,F). The expression levels of TaPBF and TaSPA followed similar patterns throughout the grain-filling stage (Figure 7C,D), where their expression clearly increased at 10 DAA and remained high until 16 DAA. The expression level of TaPDIL2-1 was highest at 10 DAA in HMW-D1p and at 13 DAA in HMW-D1a (Figure 7G). Moreover, the transcript levels of all these genes were up-regulated more in HMW-D1p than HMW-D1a.

## 3. Discussion

The concentration and composition of glutenin determine the unique viscoelastic properties of wheat dough, but especially the types and numbers of HMW-GSs [7]. Superior combinations of HMW-GSs accelerate the polymerization of glutenin during grain development, thereby improving the dough properties [9,24,25]. However, the absence of individual subunits can delay and decrease glutenin polymerization [8], as shown in the present study (Figure 1 and Figure 2). To further investigate how Dx2 affects the polymerization of glutenin and the quality of gluten, we analyzed the accumulation of PBs in the endosperm, the nitrogen metabolism levels, and the expression levels of genes related to glutenin synthesis and folding during different developmental stages.

Glutenin and gliadin are the main components of PBs [18], and the size and area of PBs are related to the quality of wheat [26]. As shown in Figure 3, PBs were not observed at 10 DAA in the HMW-D1a endosperm, whereas some small PBs were distributed in the HMW-D1p endosperm. Subsequently, more abundant and larger PBs were observed in the endosperm in HMW-D1p compared with HMW-D1a. These findings were inconsistent with the accumulation of glutenin and gliadin in the early stage of grain-filling (Figure 2), which was probably attributed to the lower amount of glutenin and GMP synthesized in the HMW-D1a endosperm, leading to PBs invisible at 10 DAA and even fewer and smaller at later stage of the grain-filling. It suggests that storage proteins were synthesized and aggregated earlier in HMW-D1p with Dx2+Dy12 than HMW-D1a with Dy12, which agrees with the results reported by Gao et al. [8] and Liu et al. [24] who observed that lines with strength-associated alleles started to form large polymers several days earlier than those without. In the present study, we used wheat lines with HMW-GS variations only at the Glu-1D locus where the same protein fractions and contents were similar except for glutenin, therefore it suggests that the difference in the accumulation of PBs in the endosperm in HMW-D1a and HMW-D1p may be related to variations in the formation of glutenin and GMP caused by the absence of Dx2.

In wheat grains, protein synthesis depends mainly on the assimilation and transport of nitrogen in the vegetative organs after anthesis [27]. The GS enzyme plays an important role in the assimilation of mineral N in wheat [28]. In the present study, the GS activity levels were higher in both the flag leaves and grains in HMW-D1p than HMW-D1a during grain development (Figure 4A,B), thereby suggesting that HMW-D1p had a higher capacity for assimilating ammonium in the flag leaves and grains. Moreover, the GPT enzyme is involved in the synthesis of amino acids such as alanine, glutamine, and glutamate [15,16]. The GPT activity levels were also higher in both the flag leaves and grains in HMW-D1p than HMW-D1a (Figure 4C,D), thereby indicating that more amino acids were synthesized in HMW-D1p during grain development compared with HMW-D1a, as demonstrated by the quantitative analyses of the amino acid contents (Figure 5A,B). In addition, the maximum amino acid contents of the flag leaves and grains occurred in HMW-D1p prior to HMW-D1a. These results suggest that more amino acids were available earlier for the subsequent synthesis of the total grain proteins and the individual components in HMW-D1p than HMW-D1a, which may explain the greater accumulation of glutenin and PBs in the HMW-D1p endosperm (Figure 2 and Figure 3).

The GS1 and GS2 isozyme activity in flag leaves are positively correlated with the total protein content of wheat grains [29]. The GS2 gene co-localizes with the grain protein content quantitative trait locus in wheat [30] and it contributes greatly to accumulation of protein [28]. Overexpression of the AlaAT gene significantly increases the grain nitrogen use efficiency [16]. Higher expression levels of cysteine and thiol proteases indicate the strong degradation and mobilization of soluble proteins in the leaves [11]. In the present study, the expression levels of TaGS (TaGS1 and TaGS2) and TaAlaAT were higher in HMW-D1p than HMW-D1a, and this was also the case for the genes encoding cysteine and thiol proteases (Figure 6). The maximum expression levels of TaGS1 and TaGS2 occurred earlier in HMW-D1p than HMW-D1a. Meanwhile, our results also suggested that HMW-D1p exhibits higher glutenin content than HMW-D1a. Therefore, compared with HMW-D1a, the higher activity and gene expression levels of nitrogen metabolism enzymes in HMW-D1p may have contributed to the more rapid synthesis of glutenin in the grains.

HMW-GS genes are specifically expressed in the endosperm [6,7], so we analyzed the expression levels of genes encoding Dx2 and Dy12. The results showed that the expression level of the Dy12 gene exhibited similar patterns but it was higher in HMW-D1a than HMW-D1p, whereas the Dx2 gene was only expressed in HMW-D1p during wheat development (Appendix A). However, the total expression level of Glu-1Dx2+1Dy12 was higher in HMW-D1p and HMW-D1a (Figure 7A), which may explain the higher glutenin content in HMW-D1p. Several studies have shown that TaGAMyb, TaPBF, and TaSPA activate the expression of HMW-GS genes [10,12,13]. The differences in the accumulation of PBs among various wheat varieties may be due to variations in the specificity of different regulatory signals in the glutenin subunit promoter [31]. In the present study, the expression levels of TaGAMyb, TaPBF, and TaSPA were higher in HMW-D1a without Dx2, thereby supporting the glutenin and PB accumulation results (Figure 2D and Figure 3). We also found that the expression patterns of TaGAMyb, TaPBF, and TaSPA were similar to that of Glu-1Dx2+1Dy12 (Figure 7B‒D), thereby indicating these three TFs up-regulated the expression of HMW-GS genes, as also indicated by Ravel et al. [12]. These findings suggest that the greater accumulation of glutenin and PBs can be attributed to the higher expression of Glu-1Dx2+1Dy12 due to up-regulation by the three TFs.

Several studies have suggested that PPIase, SUMO1, and PDI are required for gluten protein-folding and assembly during grain development [19,32]. PPIase facilitates the accumulation of protein polymers with the assistance of SUMO1 [21]. The expression of PDI catalyzes the formation of disulfide bonds, which are positively correlated with protein polymer formation [11,22]. In the present study, the expression levels of TaPPIase, TaSUMO1, and TaPDIL2-1 were higher in HMW-D1p than HMW-D1a (Figure 7E–G), and the maximum TaPDIL2-1 expression level occurred earlier in HMW-D1p, thereby suggesting that larger protein polymers were formed in HMW-D1p, as demonstrated by the accumulation of PBs and GMP, and the concentration of disulfide bonds (Figure 1A, Figure 2E, and Figure 3). Moreover, SUMO1 contributes to the stabilization of proteins during protein aggregation [33]. The disulfide bond is an important functional bond in the tertiary structure of proteins and it is closely related to the structural stability [34]. In the present study, we showed that HMW-D1p gluten exhibited a stronger cross-linking network and higher thermal stability compared with HMW-D1a (Figure 1). Therefore, we suggest that differences in the expression levels of TaPDIL2-1, TaSUMO1, and TaPPIase may explain the differences in the formation of glutenin polymers and gluten structures in HMW-D1a and HMW-D1p.

## 4. Materials and Methods

### 4.1. Plant Materials

Two wheat NILs were used with HMW-GS Dx2 absent or present at the Glu-D1 locus (designed as HMW-D1a and HMW-D1p, respectively), which were obtained from a cross between Xinong 2208 (Ax1, Bx7+By9, Dx2+Dy12) and Fa 746 (null, Bx17+By18, Dx3+Dy12) using the method described by Gao et. [8]. The glutenin and gliadin contents and compositions of BC_6_F_11_ homozygous NILs have also been confirmed [8]. HMW-D1a contains the Ax1, Bx7, By9, and Dy12 subunits, whereas HMW-D1p contains the Ax1, Bx7, By9, Dx2, and Dy12 subunits.

HMW-D1a and HMW-D1p were manually planted individually at a sowing density of 60 seeds per row in Yangling (108°40’E, 34°160’N), Shaanxi Province, China, during the 2017–2018 and 2018–2019 growing seasons. The block size was 20 × 2 m × 0.23 m and the row spacing was 0.23 m. A randomized complete block design was employed with three replicates for each wheat line. Spikes obtained from HMW-D1a and HMW-D1p were tagged immediately when the middle spikelet first reached anthesis in the morning. About 40 labeled spikes with flag leaves were randomly collected at 0, 4, 7, 10, 13, 16, 22, 28, and 34 days after anthesis (DAA). About half of the sampled grains and flag leaves were frozen in liquid nitrogen for 1 min and then stored at −80°C to analyze the amino acid contents, enzyme activities, and gene expression levels. The other grain samples were used to determine the protein compositions and for cytological studies. After harvesting, the grains were stored to allow post-maturation, milled using a multifunctional pulverizer (Y400, Laobenxing, Zhejiang, China), and screened through an 80-mesh grid.

### 4.2. Determination of Structural‒Thermal Properties of Gluten

Gluten samples were prepared according to Li et al. [9]. The sulfhydryl group and disulfide bonds, surface hydrophobicity, and thermal properties of gluten from HMW-D1a and HMW-D1p were determined according to previously described methods [25]. The microstructure of freshly prepared dough was observed by confocal laser scanning microscopy (FLUOVIEW FV1200, Olympus Optical Co. Ltd., Tokyo, Japan), and then the images were quantified using AngioTool64 (version 0.6a, National Cancer Institute, National Institute of Health, Maryland, USA) according to the method described by Gao et al. [8]. All analyses of each sample were conducted in triplicate.

### 4.3. Determination of Protein Compositions of Grains during Grain Development

The grain proteins were extracted according to the method described by Amagliani et al. [35] with modifications. Flour (0.5 g) was suspended in 5 mL of 5% NaCl at 20 °C and agitated for 60 min. The suspension was centrifuged at 5000 g for 15 min to obtain albumin and globulin. Next, the residue was subsequently agitated and extracted with 5 mL of 70% ethanol at 20 °C for 30 min and centrifuged at 5000 g for 15 min to obtain gliadin. Finally, the residue was agitated and extracted with 5 mL of 0.2% NaOH at 20 °C for 30 min and centrifuged at 5000 g for 15 min to obtain glutenin. Each extraction was repeated twice. Moreover, a flour sample weighing 0.05 g was suspended in 1 mL of 1.5% sodium dodecyl sulfate solution and centrifuged at 18100g for 15 min at 20 °C to extract the GMPs [22,36]. The contents of total protein and compositions were determined using the Kjeldahl method. All analyses were repeated in triplicate independently.

### 4.4. Cytological Analysis of Endosperm

The caryopses were collected at 10, 13, and 16 DAA to examine the development of PBs. Half of the endosperm samples were obtained and stored immediately in 0.1 M phosphate buffer (pH 6.8) with 4% glutaraldehyde, and then washed three times in sodium phosphate buffer (pH 6.8) for a total of 30 min (10 min each time). Samples were fixed in 0.5% osmic acid for 1.5–2.5 h and then washed three times (10 min each) with 0.1 M sodium phosphate buffer (pH 6.8). Next, the samples were dehydrated using an ethanol series comprising 30, 50, 70, 80, and 90% (10 min each), and then 100% (three times, 20 min each). Subsequently, white resin was used to infiltrate and embed the samples. The samples were then polymerized at 70 °C for 12 h in an oven. Finally, the samples were cut into 1-µm slices using a histotome (RM2265, Leica Microsystem Ltd., Wetzlarcity, Germany), stained with 0.03 M toluidine blue for 25 s, and observed and photographed using a Leica microscope (DMLS, Leica Microsystem Ltd., Wetzlar, Germany).

### 4.5. Activities of Enzymes Related to Protein Synthesis and Amino Acid Levels in Grains and Flag Leaves during Grain Development

The GS from fresh grains and leaves were extracted and assayed as described previously method [15]. Briefly, the fresh samples were ground using a chilled mortar in 3 mL of 5 mM sodium phosphate buffer (pH 7.2) containing 50 mM Na_2_SO_4_ and 0.5 mM Na_2_EDTA, then centrifuged at 20,000 g for 20 min at 4 °C. The reaction mixture contained 1.0 mL of extracting solution, 0.3 mL of 0.3 M sodium glutamate, 0.8 mL of 0.25 M imidazole-HCl (pH 7.0), 0.2 mL of 0.5 M MgSO_4_, and 0.4 mL of 0.03 M Na-adenosine triphosphate (pH 7.0). The mixtures were incubated at 25 °C for 5 min and then added 0.2 mL of 1.0 M hydroxylamine. After incubation at 25 °C for 15 min, the reaction was terminated using 0.8 mL mixed reagent (10% FeCl_3_.6H_2_O, 50% (v/v) HCl and 24% (w/v) trichloracetic acid). After 20 min, the mixtures were centrifuged at 15,000 g for 10 min, and then the supernatant was measured at 540 nm using an ultraviolet-visible spectrophotometer (SP-756PC, Shanghai Spectrum Instrument Co. Ltd., Shanghai, China).

The fresh grain and leaf samples (0.2 g) were ground in 4 mL of 50 mM Tris–HCl buffer (pH 7.2) using a chilled mortar, before centrifugation at 13,600× *g* for 30 min at 0 °C according to the method reported by Hu et al. [15] with some modifications. The supernatant obtained was the crude GPT enzyme extract. The crude GPT enzyme extract (0.1 mL) and GPT substrate solution (0.5 mL) containing 200 mM DL-alanine and 2 mM α-ketoglutaric acid in 0.1 M phosphate buffer (pH 7.4) were added to a 10-mL tube. Only 0.1 mL of the crude enzyme extract was added to another tube as the blank. Next, the two tubes were placed in a water bath at 37 °C and removed after 30 min. The reaction was terminated by adding 2,4-dinitrophenylhydrazine solution (0.5 mL) to each tube, but only GPT substrate solution (0.5 mL) was added to the blank tube. The tubes were again placed in a water bath at 37 °C for 20 min. Next, 5.0 mL of 0.4 M NaOH was added and the tubes were mixed. After 10 min, the absorbance of the reaction mixture was measured at 500 nm using an ultraviolet-visible spectrophotometer (SP-756PC, Shanghai Spectrum Instrument Co. Ltd., Shanghai, China).

The formula used for calculating the GS and GPT enzyme activities is:
(1)$$Enzyme\;activity\;(OD\cdot g^{-1}\cdot h^{-1}\;FW)=\frac{A\times Vt}{m\times Vs\times T}$$
where A is the optical density (OD) value, Vt is the total crude enzyme protein content (mL), m is the sample weight (g), T is the reaction time (h), vs. is the crude enzyme volume during the reaction (mL), and FW is fresh weight. All analyses were conducted in triplicate.

Free amino acids were extracted from fresh leaf and grain samples (0.2 g) with 10% acetic acid and determined using the ninhydrin method [15]. The results were expressed as *mg*/*g* FW. All analyses were conducted in triplicate.

### 4.6. Gene Expression Analysis during Grain Development

Total ribose nucleic acid (RNA) was extracted from about 80–120 mg of the HMW-D1a and HMW-D1p leaves and grains at different stages (0, 4, 7, 10, 13, 16, 22, and 28 DAA) using TRIzol reagent according to the manufacturer’s instructions (Takara, Dalian, Liaoning, China). First-strand complementary deoxyribonucleic acid (cDNA) was synthesized with purified RNA and oligo (dT15) primers according to the manufacturer’s instructions (Takara, Tokyo, Japan). The cDNA was diluted to 100 ng/μL with RNase-free double-distilled H_2_O and then stored at −20 °C until analysis.

Quantitative real-time polymerase chain reaction (qRT-PCR) was performed to analyze the transcript levels of protein synthesis-related genes and TFs using a Light Cycler^®^ 96 detection system (Roche, Basel, Switzerland) according to the manufacturer’s instructions. The primers for the genes analyzed in this study are listed in Appendix A. The reaction mixture contained 10 µL of 2× Fast-start Essential DNA Green Master (Roche, Basel, Switzerland), 0.8 µL of each primer (10 mM), 80 ng cDNA template, and double-distilled H_2_O to make up a final volume of 20 µL. The same thermal profile was used for all of the PCR reactions, i.e., 95 °C for 10 min, followed by 45 cycles at 95 °C for 10 s, 57–62 °C for 10 s, and 72 °C for 20 s, and then at 72 °C for 10 min. The relative expression levels were calculated using the 2^−∆∆CT^ method, except genes encoding HMW-GSs were calculated using the 2^−∆CT^ method, where the bread wheat 18S rRNA gene was used as an internal reference. Triplicate PCR reactions and three biological replicates were performed for each gene, where the HMW-D1a sample at 4 DAA was used as the reference sample for ∆∆CT.

### 4.7. Statistical Analysis

Analysis of variance and the Student’s t-test were conducted to determine significant differences between HMW-D1a and HMW-D1p, where differences were considered significant at *p* < 0.05. Figures were prepared using Sigmaplot 12.5 and Adobe Photoshop 7.0.

## 5. Conclusions

In this study, we investigated the effects of the absence of Dx2 on the gluten quality of wheat based on cytological, physicochemical, and transcriptional analyses using two NILs with HMW-GS Dx2 absent or present at the Glu-D1 locus. The absence of Dx2 decreased the accumulation of glutenin, GMP, and PBs. The activities and expression levels of genes involving in N metabolism enzymes were lower in HMW-D1a than HMW-D1p. Genes related to glutenin synthesis and three TFs were down-regulated in HMW-D1a without Dx2. Therefore, lower levels of nitrogen metabolism capacity and glutenin-related genes could account for the lower accumulation of glutenin, GMP, and PBs, thereby weakening the structural‒thermal properties of gluten. These findings might improve our understanding of the mechanism that allows HMW-GS Dx2 absence to affect the quality of wheat.

## Figures and Tables

**Figure 1 ijms-21-01383-f001:**
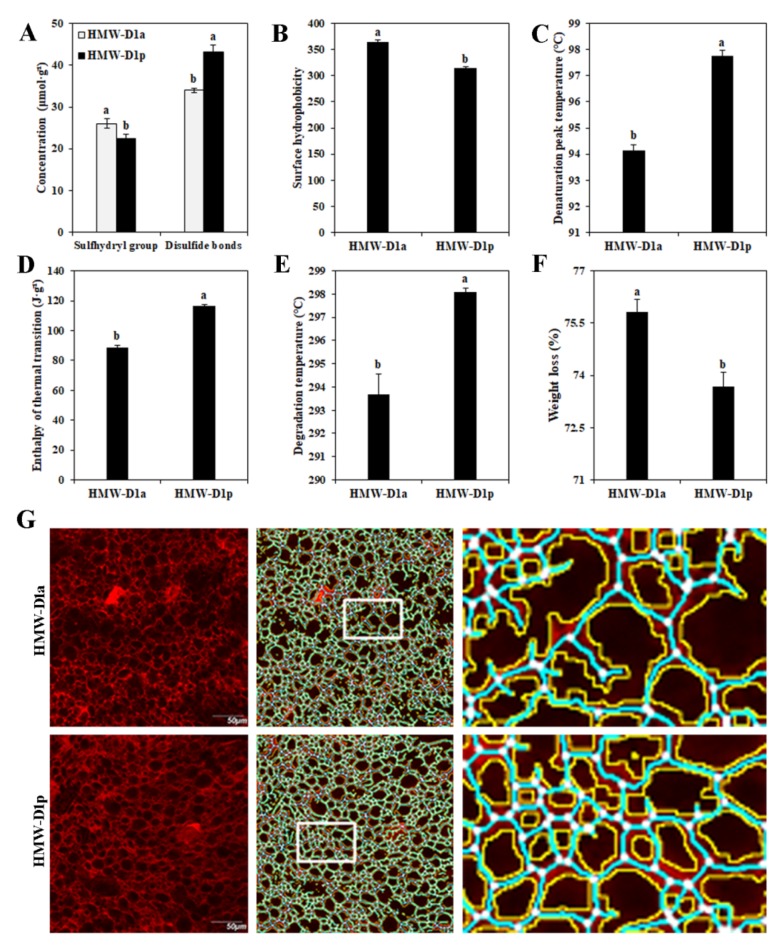
Structural‒thermal characteristics of gluten from HMW-D1a and HMW-D1p in 2017–2018 growing season. (**A**) Concentrations of sulfhydryl groups and disulfide bonds in gluten; (**B**) Surface hydrophobicity of gluten; (**C**–**F**) Thermal properties of gluten; G, Dough microstructures, the images in the left column are the original images and the scale bar represents 50 μm. The images in the middle column were processed with AngioTool. The images in the right column show the enlarged areas selected from the processed images highlighted by the white boxes, where the junctions are shown in white, the protein skeleton in blue, and the protein outline/area in yellow. Results followed by a different letter in the same column within the same group are significantly different (*p* < 0.05).

**Figure 2 ijms-21-01383-f002:**
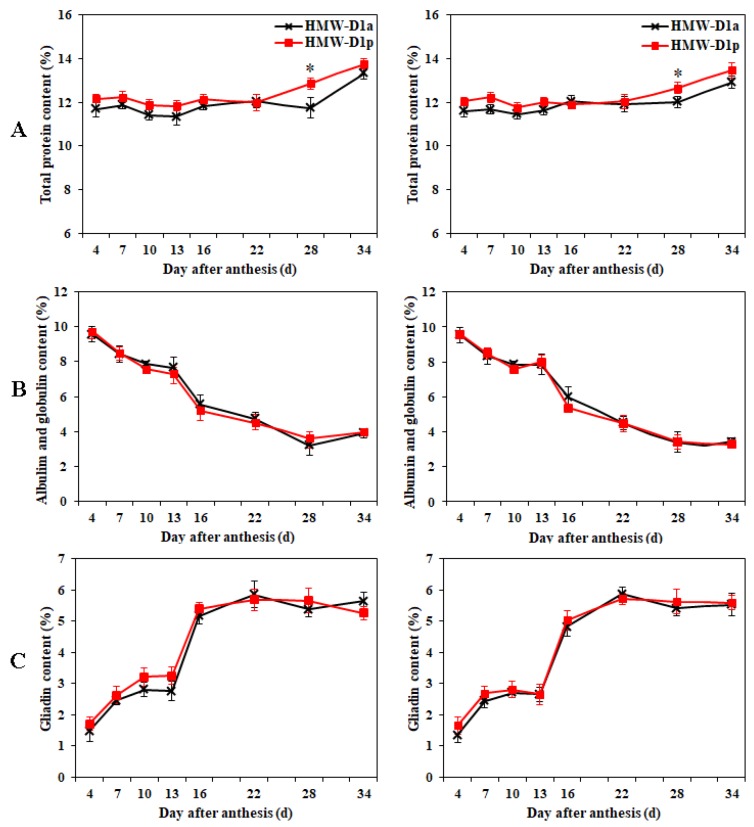
Dynamic accumulation of total protein and its components in grains from HMW-D1a and HMW-D1p over two growing seasons, 2017–2018 (left) and 2018–2019 (right). (**A**) total protein; (**B**), albumin and globulin; (**C**) gliadin; (**D**) glutenin; (**E**) GMP. Red and black lines represent the HMW-D1p and HMW-D1a, respectively. * indicates significant difference at same developing stage of the two materials (*p* < 0.05).

**Figure 3 ijms-21-01383-f003:**
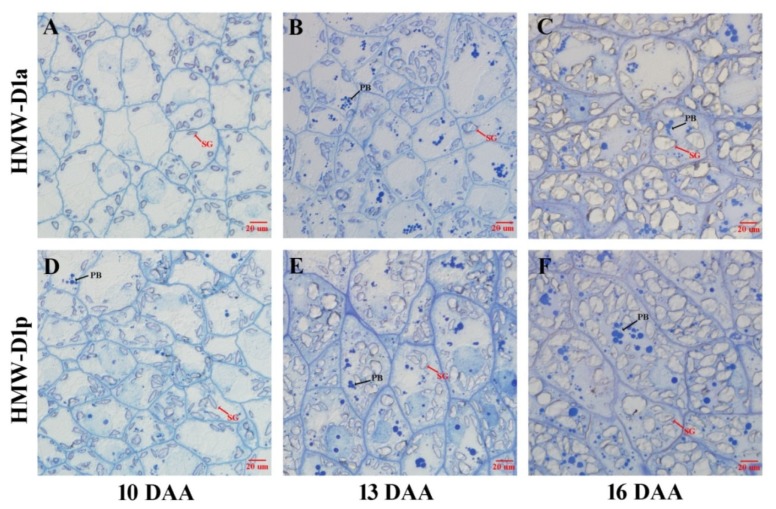
Cytological observations of wheat endosperm. (**A**–**F**) represents the endosperm at 10, 13, and 16 days after anthesis (DAA) in HMW-D1a (**A–C**) and HMW-D1p (**D–F**) with the scale bar of 20 µm. Red arrows indicate starch granules (SG) and black arrows indicate protein body (PB).

**Figure 4 ijms-21-01383-f004:**
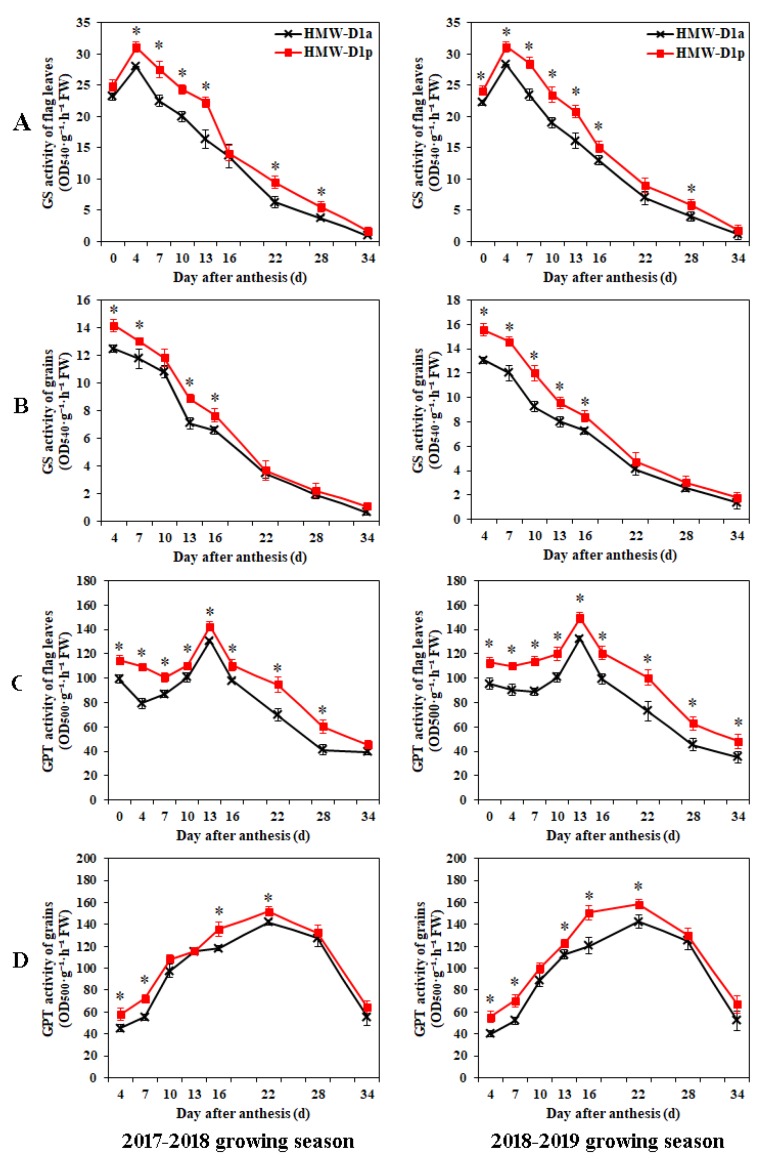
Changes in the GS and GPT in leaves (**A**,**C**) and grains (**B**,**D**) from HMW-D1a and HMW-D1p over two growing seasons, 2017–2018 (left) and 2018–2019 (right). GS, glutamine synthetase; GPT, glutamate pyruvate transaminase; FW, fresh weight. Red and black lines represent the HMW-D1p and HMW-D1a, respectively. * indicates significant difference at same developing stage of the two materials (*p* < 0.05).

**Figure 5 ijms-21-01383-f005:**
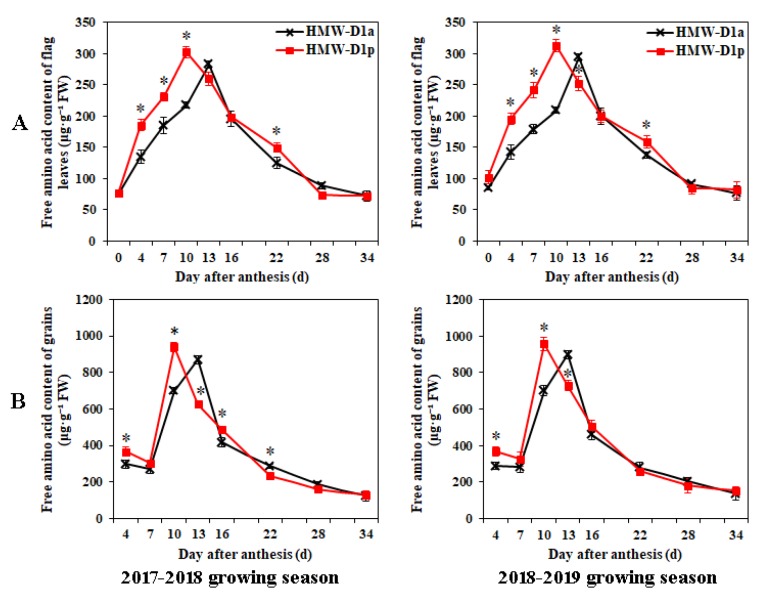
Content of total free amino acids in flag leaves (**A**) and grains (**B**) from HMW-D1a and HMW-D1p over two growing seasons, 2017–2018 (left) and 2018–2019 (right). Red and black lines represent the HMW-D1p and HMW-D1a, respectively. * indicates significant difference at same developing stage of the two materials (*p* < 0.05).

**Figure 6 ijms-21-01383-f006:**
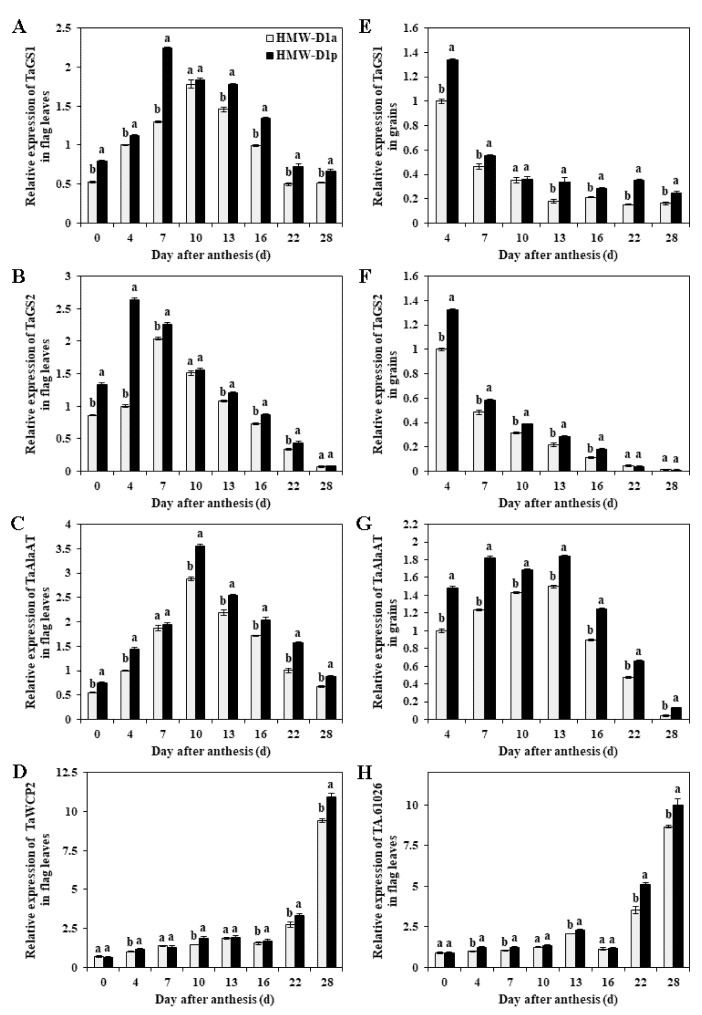
Expression levels of genes related to nitrogen metabolisms in flag leaves (**A–D**,**H**) and grains (**E–G**) from HMW-D1a and HMW-D1p in 2018–2019 growing season. (**A**,**E**), TaGS1; (**B**,**F**) TaGS2; (**C**,**G**) TaAlaAT; (**D**) TaWCP2 encoding cysteine protease; (**H**), TA.61026 encoding thiol protease. Gray column represents HMW-D1a and black column represents HMW-D1p. Results followed by a different letter in the same column within the same group are significantly different (*p* < 0.05).

**Figure 7 ijms-21-01383-f007:**
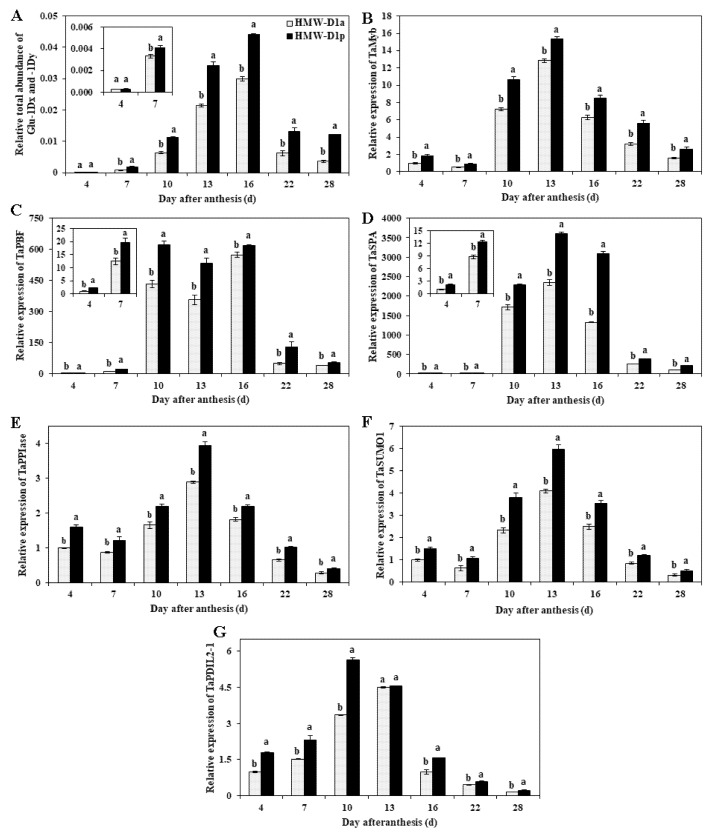
Expression levels of glutenin synthesis‒related genes in grains from HMW-D1a and HMW-D1p in 2018–2019 growing season. (**A**), Glu-1Dx+1Dy; (**B**) TaGAMyb; (**C**) TaPBF; (**D**) TaSPA; (**E**) TaPPIase; (**F**) TaSUMO1; (**G**) TaPDIL2-1. Gray column represents HMW-D1a and black column represents HMW-D1p. Results followed by a different letter in the same column within the same group are significantly different (*p* < 0.05).

**Table 1 ijms-21-01383-t001:** Parameters analyzed using AngioTool for the protein network in dough obtained from the HMW-D1a and HMW-D1p flour.

Line	Protein Area (× 10^4^ μm^2^)	Protein Junctions (× 10^2^)	Junction Density (×10^−3^)	Total Protein Length (× 10^3^ µm)	Endpoints (× 10^2^)	Lacunarity (×10^−2^)	Branching Rate (×10^−3^)	Endpoint Rate (×10^−3^)
HMW-D1a	11.63 ± 0.15b	11.74 ± 0.15b	4.50 ± 0.05b	21.88 ± 0.25b	3.60 ± 0.21a	4.81 ± 0.07b	10.09 ± 0.04b	3.09 ± 0.22a
HMW-D1p	12.79 ± 0.05a	13.64 ± 0.19a	5.07 ± 0.04a	23.77 ± 0.24a	2.64 ± 0.19b	5.76 ± 0.06a	10.66 ± 0.15a	2.06 ± 0.16b

Values are expressed as mean ± standard deviation (*n* = 10). Results followed by a different letter in the same column are significantly different (*p* < 0.05).

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
