# Peer review of "Absence of Dx2 at Glu-D1 Locus Weakens Gluten Quality Potentially Regulated by Expression of Nitrogen Metabolism Enzymes and Glutenin-Related Genes in Wheat"

_ijms, 2020, doi:10.3390/ijms21041383_

Round 1
Reviewer 1 Report
The article “Absence of Dx2 at Glu-D1 locus weakens gluten quality potentially regulated by expression of nitrogen metabolism enzymes and glutenin-related genes in wheat” provided evidences to demonstrate how Dx2 absence affect gluten quality in wheat. It may be accepted for publication after addressing the following issues: Fig. 1G dough microstructures: as far as I can see from the images, the differences on gluten microstructure between HMW-D1a and HMW-D1p did not seem obvious, authors may add arrows something to the images for a better illustration. figure 2 and fig. 4 Do you have statistical analysis of data presented in figure 2 and fig. 4? Significance should be indicated in the figures. Where is figure 5? The data of amino acid contents is missing. How to quantify the total expression level of Glu-1Dx2+1Dy12? Authors should provide more details in methods section.
Author Response
We would like to express our sincere thanks to you for the constructive comments and questions.
Manuscript ID: ijms-703589
Article type: Article
Title: Absence of Dx2 at Glu-D1 locus weakens gluten quality potentially regulated by expression of nitrogen metabolism enzymes and glutenin-related genes in wheat
Journal: International Journal of Molecular Sciences
We would like to express our sincere thanks to you for the constructive comments and questions. Point-by-point responses to your comments are provided in the following.
Comment: Fig. 1G. Dough microstructures: as far as I can see from the images, the differences on gluten microstructure between HMW-D1a and HMW-D1p did not seem obvious, authors may add arrows something to the images for a better illustration.
Response: Thank you very much for this comment. Based on your suggestion, we have further processed Fig. 1G and added Table 1 that is used to quantify the protein network structure of dough. Corresponding state has been added in line 104-107 and 114-118.
Comment: Figure 2 and Fig. 4 Do you have statistical analysis of data presented in figure 2 and fig. 4? Significance should be indicated in the figures.
Response: Yes, thanks for this comment. We have statistically analyzed the data presented in Fig. 2 and Fig. 4 according to your suggestion. The Fig. 2, Fig. 4 and their captions have been revised and reorganized. And the significance of the data has been indicated in the figures at P < 0.05 level. (Line 137, 173)
Comment: Where is figure 5? The data of amino acid contents is missing.
Response: Thank you for this comment. Due to my carelessness, the Fig. 5 was missed during the writing of the manuscript. We have added the Fig. 5 and its caption. (Line 175-179)
Comment: How to quantify the total expression level of Glu-1Dx2+1Dy12?
Response: Thank you very much for this comment. In this study, the abundance of transcripts from Glu-1Dx2 and Glu-1Dy12 were measured using the 2−ΔCT method, where the bread wheat 18S rRNA gene was used as an internal reference. The total expression level of Glu-1Dx2+1Dy12 was measured as the sum of expression abundance of Glu-1Dx2 and Glu-1Dy12.
Comment: Authors should provide more details in methods section.
Response: Thanks for this comment. Based on your suggestion, we have added more details about methods in this study, which were displayed in line 313-315, 332-336, 339-344, 352-357, and 363-372.
Reviewer 2 Report
In the manuscript “Absence of Dx2 at Glu-D1 locus weakens gluten quality potentially regulated by expression of nitrogen metabolism enzymes and glutenin-related genes in wheat” Song and co-authors studied the effects of absence of Dx2 glutenin subinut on physiochemical establishment of wheat quality. They found that the absence of Dx2 decreased the accumulation of protein bodies, activities of some key enzymes, and the expression levels of genes related to nitrogen metabolism.
The work is of good quality. However, some experiments, such as anlysis of polymerization index and molecular features of aggregates, could be done in the future to explain some of their findings.
My remarks concern:
Line 120: Results showed an increase of glutenin relative content. Conclusion should not be done on the polymerization phenomenon!
Fig 2C: any explanation for the shutdown of the gliadin fraction synthesis!?
Fig 2D: please use the same scale for 2017-2018 and 2018-2019 growing season.
Line 133: The rate of storage protein accumulation should be calculated from fig 2 C and D. Then conclusion should be done.
Fig.3 any information (changes) on the number and size of cells in the endosperm?
Line 148: Fig 5 is missing
Line 153: caption should contains all information. Example: HMW-D1a (gray line) and HMW-D1p (black line).
Line 172: there is no direct relationship between capacity for nitrogen metabolism and absence of Dx2…authors have to argue
Line 204: I am not convinced of the explanation for the absence of PBs, for both NILs lines, the accumulation of storage proteins is almost identical (fig. 2C, D and E).
Line 233: Authors should check the correlation between GS1, GS2 and the accumulation of protein.
Author Response
We would like to express our sincere thanks to you for the constructive comments and questions.
Manuscript ID: ijms-703589
Article type: Article
Title: Absence of Dx2 at Glu-D1 locus weakens gluten quality potentially regulated by expression of nitrogen metabolism enzymes and glutenin-related genes in wheat
Journal: International Journal of Molecular Sciences
We would like to express our sincere thanks to you for the constructive comments and questions. Point-by-point responses to your comments are provided in the following.
Comment: Comment: Line 120: Results showed an increase of glutenin relative content. Conclusion should not be done on the polymerization phenomenon!
Response: Thank you very much for this comment. According to you suggestion, we have rewritten the results of glutenin and GMP (Line 128-131). The sentence has revised as “The fastest accumulation of GMP was also gained from 22 to 34 DAA, but the increase rate of GMP in HMW-D1p is higher than that in HMW-D1a (Fig. 2E).” Therefore, we concluded that “The results suggest that the absence of Dx2 delayed and decreased the polymerization of glutenin.”
Comment: Fig 2C: any explanation for the shutdown of the gliadin fraction synthesis!?
Response: Thanks for your suggestion. It was important for our research to be published. We firstly want to apologize to you for my negligence. Because the result of the Fig. 3 was inconsistent with the glutenin and GMP content (Fig. 2D and 2E), we measured the dynamic accumulation of grain protein and its fractions again. The new results showed that the shutdown of gliadin fraction synthesis was not existed in 2017-2018 growing season period. Therefore, the “shutdown” could be caused by our carelessness. I'd like to express my apology to you again
Comment: Fig 2D: please use the same scale for 2017-2018 and 2018-2019 growing season.
Response: Thanks for this suggestion. The vertical axis in Fig. 2D was designed as same scale for 2017-2018 and 2018-2019 growing seasons.
Comment: Line 133: The rate of storage protein accumulation should be calculated from fig 2 C and D. Then conclusion should be done.
Response: Thank you for this comment. The rate of storage protein accumulation has been calculated from Fig. 2C and 2D (Line 147-149).
Comment: Fig.3 any information (changes) on the number and size of cells in the endosperm?
Response: Thank you very much for this comment. The aim of the present study was to observe the development difference of protein bodies in the endosperm with absence or presence of Dx2, further stating the effect of Dx2 absence on the glutenin formation. However, the number and size of cells in the endosperm are important for protein and starch accumulation, thus we will analyze it in future studies.
Comment: Line 148: Fig 5 is missing
Response: Thank you for this comment. Due to my carelessness, the Fig. 5 was missed during the writing of the manuscript. We have added the Fig. 5 and its caption. (Line 175-179)
Comment: Line 153: caption should contain all information. Example: HMW-D1a (gray line) and HMW-D1p (black line).
Response: Thank you very much for this comment. The marks of HMW-D1a and HMW-D1p in the Fig. 2‒7 have added in the captions, which were displayed in line 137-138, 154-155, 172-174, 175-179, 200-202, and 220-222.
Comment: Line 172: there is no direct relationship between capacity for nitrogen metabolism and absence of Dx2… authors have to argue
Response: Thank you very much for this comment. Although HMW-D1a without Dx2 possessed lower capacity for nitrogen metabolism, suggesting that lower nitrogen appearing as amino acids were transported to synthesize grain protein, leading to lower glutenin. However, the relationship between capacity for nitrogen metabolism and absence of Dx2 still need to be explored in future. Therefore, after a serious discussion with my team, the sentence has been revised as “These results indicate that the capacity for nitrogen metabolism was lower in HMW-D1a than HMW-D1p”. (Line 197)
Comment: Line 204: I am not convinced of the explanation for the absence of PBs, for both NIL lines, the accumulation of storage proteins is almost identical (fig. 2C, D and E).
Response: Thank you very much for this valuable feedback. It is very important for our research results to publish. After the first submission of the manuscript, one of our teams discovered this problem that Fig. 2 did not support the result of Fig. 3. Therefore, we measured the dynamic accumulation of grain protein again. The new results have been recognized in Fig. 2. The new Fig. 2D showed that compared with HMW-D1p, the glutenin contents were lower significantly in HMW-D1a, which may result in protein bodies invisible and even fewer and smaller during grain filling. (Line 236)
Comment: Line 233: Authors should check the correlation between GS1, GS2 and the accumulation of protein.
Response: Thank you for your suggestion. We check carefully the correlation between GS1, GS2 and the accumulation of protein. Thus, we have rewritten some sentences, which were exhibited in line 265-267.

Reviewer 3 Report
The manuscript by Song et al. reports that the absence of Dx2 at Glu-D1 locus weakens gluten quality, affects activities of various enzymes, and the expression levels of genes related to nitrogen metabolism. Thus the manuscript contains a set of novel interesting data. However I have a number of concerns:
- The title should be rephrased;
- In the abstract it is not clear that HMW-D1a and HMW-D1p mean wheat NILs. I think that a short definition of HMW-D1a and HMW-D1p presented within the beginning of the Results would be also helpful for the reader;
- Fig. 2&7: A, B, C etc should be described within the figure legend;
- Type of staining should be presented within the legend to Fig. 3;
- Fig. 5 is absent in the version I have;
- In the paragraph 2.5 the authors in particular describe their studies of the expression of ‘cysteine and thiol proteases’. In the appropriate part of the Discussion the authors are citing the paper by Zong et al. (BMC Plant Biol. 2018, 18, 353‒366). Nonetheless, I would like to point authors’ attention on the fact that recently classification of wheat degradome has been published (Int J Mol Sci. 2018 Dec 11;19(12). pii: E3991) showing hundreds of cysteine or thyol (this mean the same) proteases encoded in wheat genome. The exact enzymes should be defined as well as their known functions in the degradation and mobilization of soluble proteins.
Author Response
We would like to express our sincere thanks to you for the constructive comments and questions.
Manuscript ID: ijms-703589
Article type: Article
Title: Absence of Dx2 at Glu-D1 locus weakens gluten quality potentially regulated by expression of nitrogen metabolism enzymes and glutenin-related genes in wheat
Journal: International Journal of Molecular Sciences
We would like to express our sincere thanks to you for the constructive comments and questions. Point-by-point responses to your comments are provided in the following.
Comment: - The title should be rephrased;
Response: Thanks very much for your comment. Absence of Dx2 weakens quality of wheat gluten and decreased the glutenin polymerization. But it is not clear how the absence of Dx2 has these effects. Thus, we investigated the gluten quality in terms of cytological, physicochemical, and transcriptional characteristics using two near-isogenic lines with Dx2 absent or present at Glu-D1 locus. Thus, the title “Absence of Dx2 at Glu-D1 locus weakens gluten quality potentially regulated by expression of nitrogen metabolism enzymes and glutenin-related genes in wheat” was relatively appropriate and comprehensive.
Comment: - In the abstract it is not clear that HMW-D1a and HMW-D1p mean wheat NILs. I think that a short definition of HMW-D1a and HMW-D1p presented within the beginning of the Results would be also helpful for the reader;
Response: Thank you very much for this comment. According to your suggestion, we have added some details about the creation and confirmation of NILs, but we revised it in Materials and Methods. (Line 313-315)
Comment: - Fig. 2&7: A, B, C etc. should be described within the figure legend;
Response: Thank you very much for this comment. The A, B, C etc. in the Fig. 2 and 7 have been described in the figure legend. (Line 135-136, 220-221)
Comment: - Type of staining should be presented within the legend to Fig. 3;
Response: Thank you very much for this comment. The type of staining has been added in legend of Fig. 3 (Line 154-155).
Comment: - Fig. 5 is absent in the version I have;
Response: Thank you for this comment. Due to my carelessness, the Fig. 5 was missed during the writing of the manuscript. We have added the Fig. 5 and its caption. (Line 175-179)
Comment: - In the paragraph 2.5 the authors in particular describe their studies of the expression of ‘cysteine and thiol proteases’. In the appropriate part of the Discussion the authors are citing the paper by Zong et al. (BMC Plant Biol. 2018, 18, 353‒366). Nonetheless, I would like to point authors’ attention on the fact that recently classification of wheat degradome has been published (Int J Mol Sci. 2018 Dec 11;19(12). pii: E3991) showing hundreds of cysteine or thyol (this mean the same) proteases encoded in wheat genome. The exact enzymes should be defined as well as their known functions in the degradation and mobilization of soluble proteins.
Response: Thank you very much for this comment. Due to our negligence, the names of the genes encoding cysteine and thiol proteases are not correct to the specific gene. Considering that there are hundreds of genes to encode cysteine and thiol proteases, therefore, based on the genes used in the present study, we revised the names of cysteine protease as “TaWCP2” (accession number: AB109216), and thiol protease as “TA.61026” (AY253445). Corresponding revision was exhibited in line 181-182, 190, 201, and Supplement Table S1.